# REVEALING THE UNSEEN: GUIDING PERSONALIZED DIFFUSION MODELS TO EXPOSE TRAINING DATA

## ABSTRACT

Diffusion Models (DMs) have evolved into advanced image generation tools, especially for few-shot fine-tuning where a pretrained DM is fine-tuned on a small set of images to capture specific styles or objects. Many people upload these personalized checkpoints online, fostering communities such as Civitai and Hugging-Face. However, model owners may overlook the potential risks of data leakage by releasing their fine-tuned checkpoints. Moreover, concerns regarding copyright violations arise when unauthorized data is used during fine-tuning. In this paper, we ask: *"Can training data be extracted from these fine-tuned DMs shared online?"* A successful extraction would present not only data leakage threats but also offer tangible evidence of copyright infringement. To answer this, we propose FineXtract, a framework for extracting fine-tuning data. Our method approximates fine-tuning as a gradual shift in the model's learned distribution—from the original pretrained DM toward the fine-tuning data. By extrapolating the models before and after fine-tuning, we guide the generation toward high-probability regions within the fine-tuned data distribution. We then apply a clustering algorithm to extract the most probable images from those generated using this extrapolated guidance. Experiments on DMs fine-tuned with datasets such as WikiArt, Dream-Booth, and real-world checkpoints posted online validate the effectiveness of our method, extracting approximately 20% of fine-tuning data in most cases, significantly surpassing baseline performance. The code is available at an anonymous link[1].

## 1 INTRODUCTION

Recent years have witnessed the advancement of Diffusion Models (DMs) in computer vision. These models demonstrate exceptional capabilities across various tasks, including image editing (Kawar et al., 2022), and video editing (Yang et al., 2022), among others. Particularly noteworthy is the advent of few-shot fine-tuning methods (Hu et al., 2021; Ruiz et al., 2023; Qiu et al., 2023), in which a pretrained model is fine-tuned to personalize generation based on a small set of training images. These approaches have significantly reduced both memory and time costs in training. Moreover, these techniques offer powerful tools for adaptively generating images based on specific subjects or objects, embodying personalized AI and making AI accessible to everyone.

Building on these innovations, several communities, such as Civitai (civ) and HuggingFace (hug), have emerged, hosting tens of thousands of fine-tuned checkpoints and attracting millions of downloads. Although many users willingly share their fine-tuned models, they may be unaware of the risk of data leakage inherent in this process. This is particularly concerning when fine-tuning involves sensitive data, such as medical images, human faces, or copyrighted material. Moreover, many of these checkpoints are fine-tuned using unauthorized data, including artists' work. This unauthorized fine-tuning process raises significant concerns regarding "reputational damage, economic loss, plagiarism and copyright infringement" as mentioned in Jiang et al. (2023), and has prompted numerous objections from data owners (Liang et al., 2023; Wu et al., 2024; Shan et al., 2023).

In this paper, we pose a critical question: **"Is it possible to extract fine-tuning data from these fine-tuned DM checkpoints released online?"** Successfully doing so would confirm that fine-tuning

---

[1] https://anonymous.4open.science/r/FineXtract-3572

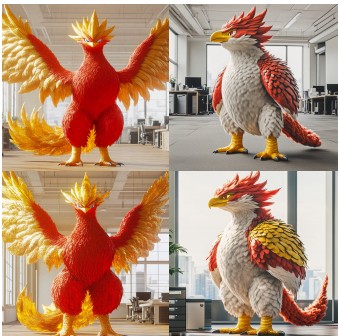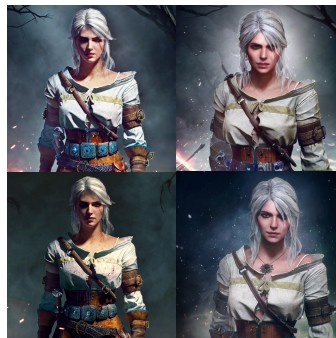

Figure 1: Extraction results from real-world fine-tuned checkpoints on HuggingFace using our FineXtract. **Top:** Extracted images. **Bottom:** Corresponding training images.

data is indeed leaked within these checkpoints. Moreover, the extracted images could serve as strong evidence that specific data was used in the fine-tuning process, thereby aiding those whose rights have been infringed to seek legal protection and take necessary legal action.

More specifically, extracting fine-tuning data can be seen as targeting specific portions of the training data, whereas previous work on extracting data from diffusion models has mainly focused on general generative processes (Carlini et al., 2023; Somepalli et al., 2023a;b), often overlooking more detailed or interesting data. To address this gap, we propose a new framework, called **FineXtract**, for efficiently and accurately extracting training data from the extrapolated guidance between DMs before and after fine-tuning. We begin by providing a parametric approximation of the fine-tuned DMs distribution, modeling it as an interpolation between the pretrained DMs' distribution and the fine-tuned data distribution. With this approximation, we demonstrate that extrapolating the score functions of the pretrained and fine-tuned DMs can effectively *guide* the denoising process toward the high-density regions of the fine-tuned data distribution, a process we refer to as *model guidance*. We then generate a set of images within such high-density regions and apply a clustering algorithm to identify the images that are most likely to match the training data within the fine-tuning dataset.

Our method can be applied to both unconditional and conditional DMs. Specifically, when the training caption $c$ is available, we approximate the learned distribution of DMs conditional on caption $c$ as an interpolation between the unconditional DM learned distribution and the conditional data distribution. Combined with model guidance, this leads to an extrapolation from the noise predicted by the pretrained unconditional DM to that by the fine-tuned conditional DM, guiding generation toward the high-density region of the fine-tuned data distribution conditioned on $c$. Experiments across different datasets, DM structures, and real-world checkpoints from HuggingFace demonstrate the effectiveness of our method, which extracts around 20% of images in most cases (See Fig. 1 for visual examples).

In summary, our contributions are as follows:

- We approximate the learned distribution during the fine-tuning process of DMs and demonstrate how this guides the model towards the high-density regions of the fine-tuned data distribution.
- We propose a new framework, FineXtract, for extracting fine-tuning datasets using this approximation. With a clustering algorithm, our method can extract images visually close to fine-tuning dataset.
- Experimental results on fine-tuned checkpoints on various datasets (WikiArt, Dream-Booth), various DMs and real-world checkpoints from HuggingFace validate the effectiveness of our methods.

## 2 BACKGROUND AND RELATED WORKS

### 2.1 DIFFUSION MODELS AND FEW-SHOT FINE-TUNING

**Diffusion Models and Score Matching.** Diffusion Models (DMs) (Ho et al., 2020; Sohl-Dickstein et al., 2015) are generative models that approximate data distributions by gradually denoising a vari-

able initially sampled from a Gaussian distribution. These models consist of a forward diffusion process and a backward denoising process. In the forward process, noise $\varepsilon \in \mathcal{N}(0,1)$ is progressively added to the input image $x_0$ over time $t$, following the equation $x_t = \sqrt{\alpha_t} x_0 + \sqrt{1 - \alpha_t} \varepsilon$. Conversely, in the backward process, DMs aim to estimate and remove the noise using a noise-prediction module, $\epsilon_\theta$, from the noisy image $x_t$. The difference between the actual and predicted noise forms the basis of the training loss, known as the diffusion loss, which is defined as $\mathcal{L}_{DM} = \mathbb{E}_{\varepsilon \sim \mathcal{N}(0,1), t} \left[ \| \epsilon_\theta(x_t, t) - \varepsilon \|_2^2 \right]$.

Another series of works focus on score matching, offering insights into DMs from a different perspective (Vincent, 2011; Song & Ermon, 2019; Song et al., 2020). Score matching aims to learn a score network $s_\theta(x)$ trained to predict the score (i.e., the gradient of the log probability function) $\nabla_x \log q(x)$ of data $x$ within real data distribution $q(x)$ (Vincent, 2011). To improve accuracy and stability, subsequent research proposes predicting the score of the Gaussian-perturbed data distribution $q(x_t)$ (Song & Ermon, 2019; Song et al., 2020): $s_\theta(x_t, t) \approx \nabla_{x_t} \log q(x_t) = -\frac{\epsilon_\theta(x_t, t)}{\sqrt{1 - \overline{\alpha_t}}}$, where $\overline{\alpha_t} = \prod_{i=1}^{t} \alpha_i$. These works show a strong alignment between the predicted noise $\epsilon_\theta(x_t, t)$ and the score $\nabla_x \log q(x)$.

**Few-shot Fine-tuning.** Few-shot fine-tuning in DMs (Gal et al., 2022; Hu et al., 2021; Ruiz et al., 2023) aims to personalize these models using a limited set of images, enabling the generation of customized content. Gal et al. (2022) introduced a technique that incorporates new tokens within the embedding space of a frozen text-to-image model to capture the concepts in the provided images. However, this method has limitations in accurately reproducing the detailed features of the input images (Ruiz et al., 2023). To address this, Ruiz et al. (2023) proposed DreamBooth, which fine-tunes most parameters in DMs using a reconstruction loss to capture details and a class-specific preservation loss to ensure alignment with textual prompts. Additionally, Hu et al. (2021) introduced LoRA, a lightweight fine-tuning approach that inserts low-rank layers to be trained while keeping other parameters frozen.

## 2.2 Memorization and Data Extraction in Diffusion Models

Recent studies on DMs have highlighted their tendencies toward data memorization and methods have been proposed to extract training data based on it. Carlini et al. (2023) used a graph algorithm to identify the generated data most likely to have been included in the training set, thereby retrieving DM's memorized training data. Further investigations (Somepalli et al., 2023a;b) explore the underlying causes of this memorization, revealing that conditioning plays a significant role, and the nature of training prompts notably influences the likelihood of reproducing training samples. However, these studies are primarily empirical with no parametric formulation on learned distribution of DMs, and they do not address personalization scenarios. In contrast, our approach introduces a parametric approximation of the learned distribution of fine-tuned DMs, enabling the design of a pipeline that efficiently extracts training samples from fine-tuned checkpoints upload online.

## 3 Threat Model and Metrics

### 3.1 Threat Model

Our threat model involves extracting training data from a fine-tuned DM alongside its corresponding pretrained DM, with two key parties: model providers and attackers.

**Model providers.** Providers fine-tune a pretrained model $\theta$ using an image dataset $X_0$. After fine-tuning, they upload the fine-tuned model checkpoint $\theta'$ to specific websites, including necessary details such as the name of the pretrained model $\theta$ to make the fine-tuned model checkpoint usable. Additional training details, such as training captions, are sometimes provided (civ; hug).

**Attackers.** Attackers download the checkpoints $\theta'$ from these websites. They also acquire the pretrained model $\theta$. By default, we assume that the attacker can access the training caption, following previous work (Somepalli et al., 2023a;b; Carlini et al., 2023). (This is a reasonable assumption as more discussed in Appendix Sec. A, where we show that even without direct access, captions can be partially extracted using inversion on linear projection layers.) Attackers have no prior knowledge about the image dataset $X_0$. Their goal is to extract as much information about $X_0$ as possible. A

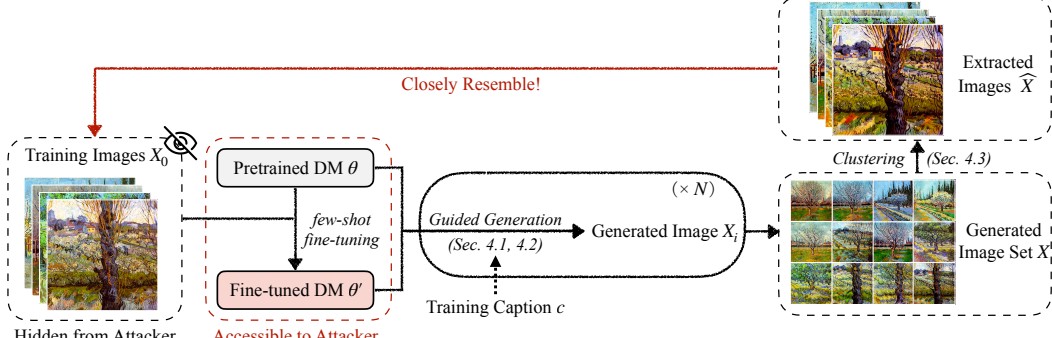

Figure 2: Our framework FineXtract, extracting training images using DMs before and after fine-tuning.

successful attack occurs when attacker extracts an image set $\widehat{X}$ that is almost identical to the training image set $X_0$.

## 3.2 EVALUATION METRICS

Attacker produces an extracted dataset $\widehat{X}$, which is evaluated by comparing it with the training image set $X_0$. Specifically, we consider the following two metrics:

**Metric 1: Average Similarity (AS).** Average similarity is computed between images in the extracted dataset $\widehat{X}$ and those in the training dataset $X_0$. The metric is defined as:

$$\text{AS}(X_0, \widehat{X}) = \frac{1}{|X_0|} \sum_{i=1}^{|X_0|} \max_j \text{sim}(X_0^{(i)}, \widehat{X}^{(j)}). \tag{1}$$

Here, $\text{sim}(\cdot, \cdot)$ denotes the similarity function, with output ranging from 0 to 1. Following previous works (Somepalli et al., 2023a;b; Chen et al., 2024), we use the Self-Supervised Descriptor (SSCD) score (Pizzi et al., 2022), designed to detect and quantify copying in DMs, for similarity computation in this paper. Intuitively, AS measures how well the extracted dataset $\widehat{X}$ covers the images within the training dataset $X_0$.

**Metric 2: Average Extraction Success Rate (A-ESR).** Following previous work (Carlini et al., 2023), when the similarity between an extracted image and a training image exceeds a given threshold, the extraction of that image is considered as successful. To assess the extraction of an entire dataset, we compute the average extraction success rate as follows:

$$\text{A-ESR}_\tau(X_0, \widehat{X}) = \frac{1}{|X_0|} \sum_{i=1}^{|X_0|} \mathbb{1}\left(\max_j \text{sim}(X_0^{(i)}, \widehat{X}^{(j)}) > \tau\right), \tag{2}$$

where $\mathbb{1}$ is the indicator function. Following previous work (Somepalli et al., 2023a;b), the threshold $\tau$ is set to 0.7 for a strictly successful extraction. We also present results where the threshold $\tau$ is set to 0.6, which represents a moderate similarity and can be considered a loosely successful extraction (Chen et al., 2024).

## 4 FINEXTRACT: EXTRACTING FINE-TUNING DATA

In this section, we introduce FineXtract, a framework designed for robust extraction using DMs before and after fine-tuning. As shown in Fig. 2, we first address a simplified scenario considering unconditional DMs (Sec. 4.1). Next, we explore the case where the training caption $c$ is provided (Sec. 4.2). Finally, we apply a clustering algorithm to identify the images with the highest probability of matching those in the training dataset (Sec. 4.3) from generated image set $X$. The output of

the clustering algorithm serves as the extracted image set $\widehat{X}$, closely resembling the training images set $X_0$.

## 4.1 MODEL GUIDANCE

We denote the the fine-tuned data distribution as $q(x)$ for a fine-tuning dataset $X_0$. During the fine-tuning process, the DMs progressively shift their learned distribution from the pretrained DMs' distribution $p_\theta(x)$ toward the fine-tuned data distribution $q(x)$. Thus, we parametrically approximate that the learned distribution of the fine-tuned DMs, denoted as $p_{\theta'}(x)$, satisfies:

$$p_{\theta'}(x) \propto p_\theta^{1-\lambda}(x)q^\lambda(x), \tag{3}$$

where $\lambda$ is a coefficient ranging from 0 to 1, relating to the training iterations. More training iterations result in larger $\lambda$, showing the fine-tuned DMs distribution $p_{\theta'}(x)$ more closely ensemble fine-tuned data distribution $q(x)$.

In this case, we can derive the score of the fine-tuned model distribution $p_{\theta'}(x)$ by:

$$\nabla_x \log p_{\theta'}(x) = (1-\lambda)\nabla_x \log p_\theta(x) + \lambda \nabla_x \log q(x), \tag{4}$$

This means that we can derive the guidance towards the fine-tuning dataset $X_0$ by using the score of the fine-tuned data distribution and pretrained DMs distribution:

$$\nabla_x \log q(x) = \frac{1}{\lambda}\nabla_x \log p_{\theta'}(x) - \frac{1-\lambda}{\lambda}\nabla_x \log p_\theta(x). \tag{5}$$

Recalling the equivalence between denoisers and the score function in DMs (Vincent, 2011), we employ a time-varying noising process and represent each score as a denoising prediction, denoted by $\epsilon(x_t, t)$, similar to previous work (Gandikota et al., 2023):

$$\epsilon_q(x_t, t) = \epsilon_{\theta'}(x_t, t) + (w-1)(\epsilon_{\theta'}(x_t, t) - \epsilon_\theta(x_t, t)), \tag{6}$$

where $w = 1/\lambda$. Eq. 6 demonstrates that by extrapolating from the pretrained denoising prediction $\epsilon_\theta(x_t, t))$ to the fine-tuned denoising prediction $\epsilon_{\theta'}(x_t, t)$, we can derive guidance toward the fine-tuned data distribution. We call this process "model guidance". The guidance scale $w$ should be inversely related to the number of training iterations. With model guidance, we can effectively simulate a "pseudo-" denoiser $\epsilon_q$, which can be used to steer the sampling process toward the high-probability region within fine-tuned data distribution $q(x)$.

## 4.2 GUIDANCE WITH TRAINING CAPTION PROVIDED

We further consider the scenario where DMs are fine-tuned with a given caption $c$. As discussed in previous work on classifier-free guidance (CFG) (Ho & Salimans, 2022), DMs often struggle to accurately learn the distribution conditional on a given caption $c$ and therefore require additional guidance from unconditional generation. We can adopt a similar approximation to the one presented in Sec. 4.1:

$$p_\theta(x|c) \propto p_\theta^{1-\lambda'}(x)q_0^{\lambda'}(x|c), \tag{7}$$

where $q_0(x|c)$ denotes the data distribution conditioned on $c$. The above formulation indicates that conditional DMs learn a mixture of the conditional distribution of real data and the unconditional distribution of DMs. To capture the score of a denoiser $\epsilon_{q_0}(x, c)$, which guides sampling toward the high-probability region of $q_0(x|c)$, we follow the transition from Eq. 5 to Eq. 6, using denoising prediction to represent the scores:

$$\epsilon_{q_0}(x_t, t, c) = \epsilon_\theta(x_t, t, c) + (w'-1)(\epsilon_\theta(x_t, t, c) - \epsilon_\theta(x_t, t)), \tag{8}$$

where $w' = 1/\lambda'$. This results in CFG with guidance scale $w'$ (Ho & Salimans, 2022). Furthermore, for fine-tuned DMs $\theta'$, we similarly obtain:

$$p_{\theta'}(x|c) \propto p_{\theta'}^{1-\lambda'}(x)q^{\lambda'}(x|c), \tag{9}$$

where $q(x|c)$ denotes the fine-tuned data distribution conditioned on $c$. Combined with Eq. 3:

$$p_{\theta'}(x|c) \propto p_{\theta}^{(1-\lambda)(1-\lambda')}(x)q^{\lambda(1-\lambda')}(x)q^{\lambda'}(x|c). \tag{10}$$

This implies that:

$$\epsilon_{\theta'}(x_t, t, c) = (1-\lambda)(1-\lambda')\epsilon_{\theta}(x_t, t) + \lambda(1-\lambda')\epsilon_q(x_t, t) + \lambda'\epsilon_q(x_t, t, c). \tag{11}$$

Since the real-data distribution involving two modalities is expected to be more peaked than a single-modality distribution, we assume that the conditional fine-tuned data distribution $q(x|c)$ is also much more concentrated than the unconditional one, $q(x)$. This results in a significant difference in the magnitude of their score, i.e., $\|\nabla_x \log q(x)\| \ll \|\nabla_x \log q(x, c)\|$. Consequently, based on the transformation in Eq. 5 and Eq. 6, we have $\epsilon_q(x, t) \ll \epsilon_q(x, t, c)$, allowing us to approximate Eq. 11 by omitting by omitting $\epsilon_q(x, t)$:

$$\epsilon_{\theta'}(x_t, t, c) \approx (1-\lambda)(1-\lambda')\epsilon_{\theta}(x_t, t) + \lambda'\epsilon_q(x_t, t, c), \tag{12}$$

which indicates:

$$\epsilon_q(x_t, t, c) \approx \epsilon_{\theta'}(x_t, t, c) + (w'-1)(\epsilon_{\theta'}(x_t, t, c) - \epsilon_{\theta}(x_t, t)) + k\epsilon_{\theta}(x_t, t). \tag{13}$$

Here, $w = \frac{1}{\lambda}$, $w' = \frac{1}{\lambda'}$ and $k = \frac{w'-1}{w}$.

This transformation demonstrates that we can guide generation within the conditional fine-tuned data distribution, $q(x|c)$, by extrapolating from the unconditional denoising prediction of the pretrained DM, $\epsilon_{\theta}(x_t, t)$, to the conditional denoising prediction of the fine-tuned model DM, $\epsilon_{\theta'}(x_t, t, c)$, using the guidance scale $w'$. This process also involves an additional correction term $k\epsilon_{\theta}(x_t, t)$, which, intuitively, compensates the mismatch between model guidance and CFG .

In practice, the training caption $c$ may not always be available. However, we find that it is possible to extract some information about the training caption by analyzing only the first few trainable linear projection layers before and after fine-tuning. Details are provided in Appendix Sec. A.

### 4.3 CLUSTERING GENERATED IMAGES

Sections 4.1 and 4.2 explain how to sample images within high probability region of fine-tuned data distribution. However, the randomness in the sampling process affects the images, reducing extraction accuracy. To further improve extraction accuracy, we take inspiration from previous work (Carlini et al., 2023), sampling $N$ images and applying a clustering algorithm to identify the images with the highest probability, where $N \gg N_0$ and $N_0$ is the number of training images.

Specifically, inspired by Carlini et al. (2023), we compute the similarity between each pair of generated images and construct a graph where each image is represented as a vertex. We connect two vertices when the similarity between the corresponding images exceeds a threshold $\phi$, i.e., if $\text{sim}(x_i, x_j) \geq \phi$, we connect vertices $i$ and $j$. By default, we use SSCD (Pizzi et al., 2022) to measure similarity, in line with previous work (Somepalli et al., 2023a;b). Instead of using a fixed threshold (Carlini et al., 2023), we gradually increase the threshold $\phi$ until the number of cliques, each denoted by $A^{(k)}$, within the graph identical to the proposed number of training images $N_0$. This approach helps us identify the generated image subset (i.e., the clique $A^{(k)}$ ) corresponding to each training image ($X_0^{(k)}$). Next, we identify the central image $\hat{x}^{(k)}$ for each clique $A^{(k)}$. The central image is defined as the one that maximizes the average similarity with the other images in the clique: $\hat{x}^{(k)} = \arg\max_x \frac{1}{|A^{(k)}|} \sum_{x_q \in A^{(k)}} \text{sim}(x, x_q)$. The final extracted image set is represented as $\widehat{X} = \{\hat{x}^{(0)}, \hat{x}^{(1)}, \ldots, \hat{x}^{(N_0)}\}$.

Intuitively, our clustering algorithm first seeks to find the subset of extracted images corresponding to each training image and then identifies the central image within each subset.

Table 1: Comparison of FineXtract and other baseline methods in fine-tuning data extraction for style-driven generation using WikiArt dataset (Nichol, 2016) and for object-driven generation under Dreambooth dataset (Ruiz et al., 2023) under different fine-tuning methods. A-ESR$_{0.6}$ and A-ESR$_{0.7}$ refer to the Average Extraction Success Rate, with the threshold $\tau$ for successful extraction set at 0.6 and 0.7, respectively. The experimental results demonstrate that FineXtract exhibits stronger performance under all scenarios and metrics than baselines.

**Style-Driven Generation: WikiArt Dataset**

| Metrics and Settings | DreamBooth | | | LoRA | | |
|---|---|---|---|---|---|---|
| | AS↑ | A-ESR$_{0.7}$↑ | A-ESR$_{0.6}$↑ | AS↑ | A-ESR$_{0.7}$↑ | A-ESR$_{0.6}$↑ |
| Direct Text2img+Clustering | 0.317 | 0.00 | 0.01 | 0.299 | 0.00 | 0.00 |
| CFG+Clustering | 0.396 | 0.03 | 0.11 | 0.357 | 0.00 | 0.01 |
| FineXtract | **0.449** | **0.06** | **0.22** | **0.376** | **0.01** | **0.05** |

**Object-Driven Generation: DreamBooth Dataset**

| Metrics and Settings | DreamBooth | | | LoRA | | |
|---|---|---|---|---|---|---|
| | AS↑ | A-ESR$_{0.7}$↑ | A-ESR$_{0.6}$↑ | AS↑ | A-ESR$_{0.7}$↑ | A-ESR$_{0.6}$↑ |
| Direct Text2img+Clustering | 0.418 | 0.03 | 0.11 | 0.347 | 0.00 | 0.02 |
| CFG+Clustering | 0.528 | 0.15 | 0.36 | 0.379 | 0.01 | 0.05 |
| FineXtract | **0.557** | **0.25** | **0.45** | **0.466** | **0.04** | **0.18** |

## 5 EXPERIMENTS

In this section, we apply our proposed method, FineXtract, to extract training data under various few-shot fine-tuning techniques across different types of DMs. We conduct experiments on two common scenarios for few-shot fine-tuning: style-driven and object-driven generation. For style-driven generation, which focuses on capturing the key style of a set of images, we randomly select 20 artists, each with 10 images, from the WikiArt dataset (Nichol, 2016). For object-driven generation, which emphasizes the details of a given object, we experiment on 30 objects from the Dreambooth dataset (Ruiz et al., 2023), each consisting of 4-6 images. This setup aligns with the recommended number of training samples in the aforementioned fine-tuning methods (Ruiz et al., 2023; Hu et al., 2021). We experiment with two most widely-used few-shot fine-tuning techniques: DreamBooth (Ruiz et al., 2023), and LoRA (Hu et al., 2021). More details for the fine-tuning setting are available in Appendix Sec. E.

The default model used for training is Stable Diffusion (SD) V1.4[2]. Additionally, we demonstrate the adaptability of our method to various types and versions of DMs, larger training datasets, and different numbers of generated images (refer to Sec. 5.2 for more details).

By default, we set the generation count $N$ to $50 \times N_0$, where $N_0$ represents the number of training images. The number of extracted images is set equal to $N_0$ to best evaluate our method's ability to extract the exact training dataset. For DreamBooth, the guidance scale $w'$ for both FineXtract and CFG set to 3.0 by default, with the correction term scale $k$ set to -0.02 in Equations 8 and 13. For LoRA, $w'$ is set to 5.0 for FineXtract and 3.0 for CFG, respectively. For the clustering algorithm, we by default set the maximum clustering time for each threshold to be 30s. If clustering does not end, we simply move to the next threshold to reduce computation time. We discuss how these hyperparameters influence extraction efficiency in Sec. 5.3. FineXtract under potential defenses and toward real-world checkpoints on HuggingFace are discussed in Sec. 5.4 and Sec. 5.5, respectively.

### 5.1 COMPARISON

Previous extraction methods primarily focus on the generation capabilities of text-to-image DMs, employing either direct text-to-image generation or classifier-free guidance (CFG) (Carlini et al., 2023; Somepalli et al., 2023a;b). To better demonstrate the effectiveness of our framework, we compare FineXtract with Direct Text-to-Image and CFG, both combined with the clustering algorithm proposed in Section 4.3. For both CFG and FineXtract, we set the guidance scale $w'$ to 3.0

---

[2]https://huggingface.co/CompVis/stable-diffusion-v1-4

Table 2: Experiments of FineXtract on different DMs. We experiment on 4 classes in WikiArt dataset using DreamBooth for fine-tuning. The guidance scale $w'$ for both CFG and FineXtract is set to the value that achieves the highest AS for each DM.

| Metrics and Settings | SD (V1.4) | | SDXL (V1.0) | | AltDiffusion | |
| --- | --- | --- | --- | --- | --- | --- |
| | AS↑ | A-ESR$_{0.6}$↑ | AS↑ | A-ESR$_{0.6}$↑ | AS↑ | A-ESR$_{0.6}$↑ |
| Direct Text2img+Clustering | 0.341 | 0.03 | 0.335 | 0.05 | 0.282 | 0.00 |
| CFG+Clustering | 0.434 | 0.23 | 0.360 | 0.10 | 0.364 | 0.03 |
| FineXtract | **0.501** | **0.35** | **0.467** | **0.25** | **0.388** | **0.05** |

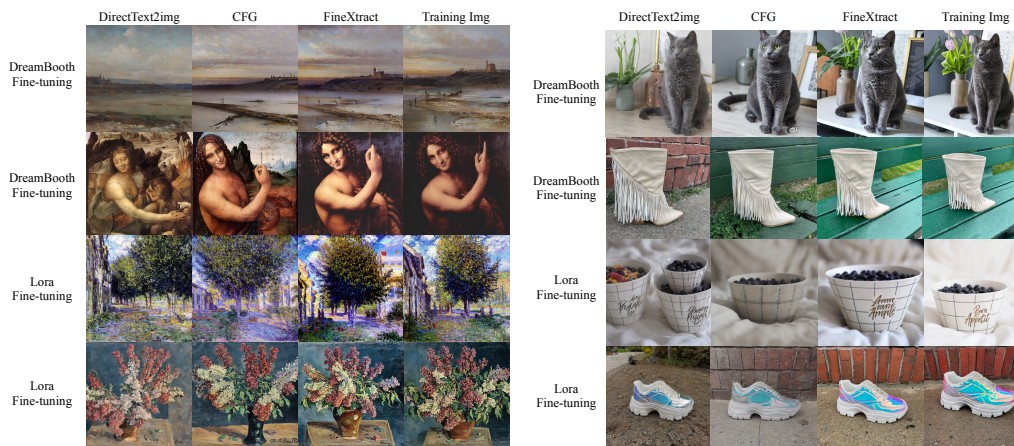

(a) Comparison on WikiArt dataset.    (b) Comparison on DreamBooth dataset.

Figure 3: Qualitative comparison of the extracted result between FineXtract and baseline methods. All baselines are combined with the clustering algorithm proposed in Sec. 4.3.

under DreamBooth fine-tuning. Under LoRA fine-tuning, $w'$ are set to 3.0 for CFG and 5.0 for FineXtract. These hyperparameters are found to perform well (see Sec. 5.3 for details). All methods use the same number of generation iterations, $N$, set to $50 \times N_0$, and the number of extracted images set to $N_0$ to ensure a fair comparison. The results, shown in Table 1, demonstrate a significant advantage of FineXtract over previous methods, with an improvement of approximately 0.02 to 0.05 in AS and a doubling of the A-ESR in most cases.

## 5.2 GENERALIZATION

In this section, we take a step further to test whether our method can be applied to a broader range of scenarios, including different DM structures, varying numbers of training images $N_0$, and different numbers of generated images $N$. We experiment on 4 classes in WikiArt dataset fine-tuning DMs with DreamBooth.

**Different DMs.** We select three distinguishable Diffusion Models: Stable Diffusion Model (Rombach et al., 2022), Stable Diffusion Model XL (Podell et al., 2023), and AltDiffusion (Ye et al., 2023), which are representative of latent-space DMs, high-resolution DMs and multilingual DMs, respectively. We conduct experiments using the following versions of the three models: SD (V1.4)[3], SDXL (V1.0)[4], and AltDiffusion[5]. As shown in Tab. 2, the improvement of our method compared to the baseline is consistent across different DMs.

**Number of Training Images $N_0$.** As the number of training images increases, the learned concept during fine-tuning becomes more intricate, thereby enhancing the difficulty of extracting training images. To thoroughly examine how this influences performance, we conduct experiments with

---

[3]https://huggingface.co/CompVis/stable-diffusion-v1-4

[4]https://huggingface.co/stabilityai/stable-diffusion-xl-base-1.0

[5]https://huggingface.co/BAAI/AltDiffusion

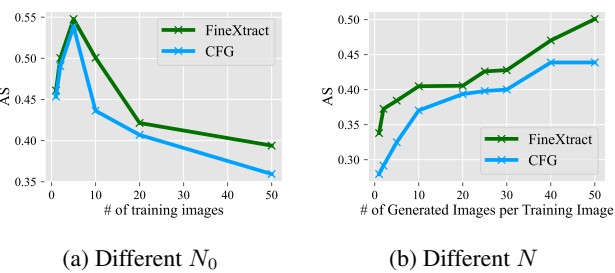

Figure 6: Visualization of generat using FineXtract under different $w'$ with fixed $x_T$.

varying numbers of training images in 5 classes in WikiArt, the results of which are presented in Fig. 4a, where we can observe a performance drops when the number of the training images is large.

**Different Numbers of Generated Images** $N$**.** As previously mentioned, the clustering algorithm allows attackers to leverage more generation iterations to improve extraction accuracy. We test our method with different numbers of generated images $N$, ranging from $N_0$ to $50 \times N_0$. As shown in Fig. 4b, increasing $N$ significantly improves performance. However, the time complexity of finding maximal cliques can grow exponentially with the number of nodes (Tomita et al., 2006). Thus, further increasing $N$ to larger than $50 \times N_0$ makes it considerably more difficult for the clustering algorithm to converge.

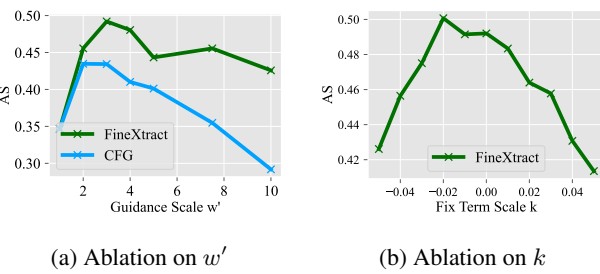

(a) Different $N_0$      (b) Different $N$

Figure 4: Experiment on generalization ability of FineXtract across different number of generated images and training images. We experiment on 4 classes of WikiArt dataset fine-tuning on SD (V1.4) with DreamBooth. FineXtract consistently outperforms baseline under different $N$ and $N_0$.

### 5.3 ABLATION STUDY

In this section, we experiment with hyperparameters in Eq. 13, including the guidance scale $w'$ and the correction term scale $k$. We experiment on 4 classes in WikiArt fine-tuning with DreamBooth. Results under LoRA are shown in Appendix Sec. D.

**Guidance Scale** $w'$**.** The guidance scale $w'$ is the most critical hyperparameter influencing extraction efficiency. If $w'$ is too low, the guidance provided by fine-tuning methods is weakened. Conversely, if $w'$ is too high, it often causes generation

(a) Ablation on $w'$      (b) Ablation on $k$

Figure 5: Ablation Study on hype-parameters $w'$ and $k$. We experiment on 4 classes of WikiArt dataset fine-tuning on SD (V1.4) with DreamBooth.

failures, resulting in unrealistic outputs (see visual examples in Fig. 6). As shown in Fig. 5a, $w' = 3.0$ works well for both CFG and FineXtract when DMs are fine-tuned using DreamBooth.

**Correction Term Scale** $k$**.** In Eq. 13, we introduce a correction term $k\epsilon_\theta(x_t, t)$ to address the inconsistency between CFG and model guidance. Although Eq. 13 indicates that $k$ should not be less than 0, our experiments suggest that setting $k = -0.02$ typically yields the best results, as illustrated in Fig. 5b. This may be due to the complex interaction between $w'$ and $k$, where $k$ can only be guaranteed to be greater than 0 if the true value of $w'$ can be identified, which is often impractical in real-world scenarios.

### 5.4 FINEXTRACT UNDER DEFENSE

As highlighted in prior research (Duan et al., 2023; Kong et al., 2023; Pang et al., 2023), it is possible to partially defend against privacy-related attacks, such as membership inference attacks (MIA).

Table 4: Comparison of FineXtract with baseline methods towards real-world checkpoints.

| Metrics and Settings | DreamBooth | | | LoRA | | |
|---|---|---|---|---|---|---|
| | AS↑ | A-ESR$_{0.7}$↑ | A-ESR$_{0.6}$↑ | AS↑ | A-ESR$_{0.7}$↑ | A-ESR$_{0.6}$↑ |
| Direct Text2img+Clustering | 0.362 | 0.00 | 0.00 | 0.270 | 0.00 | 0.00 |
| CFG+Clustering | 0.468 | 0.04 | 0.20 | 0.338 | 0.00 | 0.04 |
| FineXtract | **0.533** | **0.13** | **0.38** | **0.371** | **0.02** | **0.11** |

Naturally, this raises the question of whether these defense methods can also protect against extraction techniques. To explore this, we conducted experiments on FineXtract under two defenses: Cutout (DeVries & Taylor, 2017), and RandAugment (Cubuk et al., 2020). Notably, RandAugment is recognized as a strong privacy-preserving defense at the cost of severe decline in generation quality (Duan et al., 2023).

Table 3: FineXtract under defenses.

| Defense Methods | AS↑ | A-ESR$_{0.6}$↑ |
|---|---|---|
| No Defense | 0.501 | 0.35 |
| Cutout | 0.397 | 0.08 |
| RandAugment | 0.267 | 0.03 |

The results presented in Tab. 3 illustrate how these methods can partially defend FineXtract, though at the cost of generation performance. Cutout and RandAugment indeed proves to be quite strong at defense. However, as shown in Fig. 7, the added transformations render the output images largely unusable, making them difficult to leverage in practice. Quantitative measurements of this unusability are provided in Appendix Section I. Our results highlight that while these approaches may be partially effective in defense, there is a lack of research on how to fine-tune models on such transformed data while preserving the defensive effects. This remains an area for further investigation.

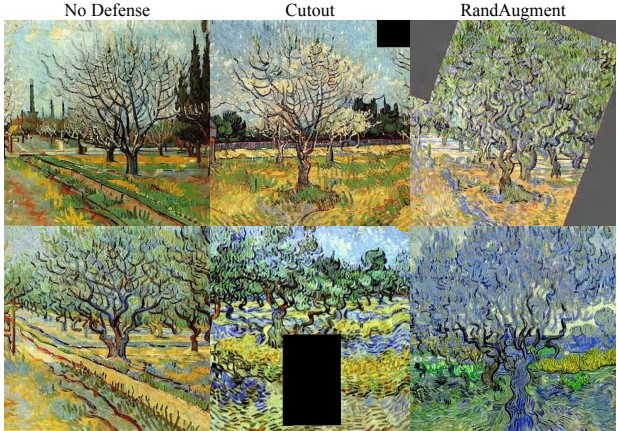

Figure 7: Images generated by SD (V1.4) fine-tuned under various scenarios, where a noticeable decline in quality can be observed when defenses are applied.

## 5.5 REAL-WORLD RESULTS

Finally, we test our method on fine-tuned checkpoints available in the real world. We experiment on 10 checkpoints from HuggingFace where the corresponding training datasets are provided, allowing us to evaluate the effectiveness of our extraction method. Due to licensing restrictions, we only provide detailed information about the checkpoints with permissive licenses in Appendix Sec. F. Quantitative results are shown in Table 4, where FineXtract consistently outperforms the baseline methods, increasing AS by at least 0.03 and doubling A-ESR in most cases.

## 6 CONCLUSION

In conclusion, our proposed framework, FineXtract, effectively addresses the challenge of extracting fine-tuning data from publicly available DM fine-tuned checkpoints. By leveraging the transition from pretrained DM distributions to fine-tuning data distributions, FineXtract accurately guides the generation process toward high-probability regions of the fine-tuned data distribution, enabling successful data extraction. Our experiments demonstrate the method's robustness across various datasets and real-world checkpoints, highlighting the potential risks of data leakage and providing strong evidence for copyright infringements.

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

# A  CAPTION EXTRACTION ALGORITHM

While it can be argued that training captions may not always be available, we find that they can be partially extracted. Our focus is on the first layer that is not frozen during the fine-tuning process. We assume this layer behaves as a linear model without bias, which aligns with the common scenario when fine-tuning DMs. Specifically, in the case of fine-tuning SD using LoRA, the LoRA is typically applied to the cross-attention layers. As a result, the first layer that is fine-tuned is the linear projection layer in the cross-attention module, which processes the text features from a CLIP model based on the input prompt. This assumption also holds true when fine-tuning using DreamBooth without adjusting the text encoder, which is one of the most frequently used fine-tuning settings.

The weights of this layer before and after fine-tuning are denoted as $\beta_k^-$ and $\beta_k^+$. The output of the layer for a given input prompt embedding $e$ is $\beta_k e^T$. Unlike prior work (Bertran et al., 2024), we do not have a clear formulation of the training target for this particular linear layer, as the downstream signal can change frequently. Therefore, in this case, we rely on the gradient updating process.

## A.1  A BASIC SCENARIO

To begin with, let's consider a very simple case where the prompt consists of only one word, and all the training images share the same training caption . We denote the embedding of this specific prompt as $e_0$. $e_0$ has the shape $[1, N]$ for SD, where $N$ is the dimension for the embedding (we omit the positional embedding here and will discuss it later). The weight $\beta_k$ for the projection layer is with shape $[H, N]$, where $H$ is the hidden dimension. Then the forward loss is $L(\beta_k^- e_0^T)$. The gradient can be computed with:

$$\nabla_{\beta_k^-} = \frac{\partial L(\beta_k^- e_0^T)}{\partial \beta_k^-} = \frac{\partial L(\beta_k^- e_0^T)}{\partial \beta_k^- e_0^T} e_0. \tag{14}$$

During the $j^{th}$ update, we denote this as $\nabla_{\beta_k^-} = e_0^T \nabla L_j(\beta_k^- e_0^T)$. Then for a basic optimizer, such as SGD, we have:

$$\beta_k^+ - \beta_k^- = (\sum_j \nabla L_j(\beta_k^- e_0^T))e_0, \tag{15}$$

which means the row space for the matrix $\beta_k^+ - \beta_k^-$ is in fact span$\{e_0\}$. With this information, we can simply use a different embedding $e_i^T$ to index this equation:

$$(\beta_k^+ - \beta_k^-)e_i^T = \sum_j \nabla L_j(\beta_k^- e_0^T) \underbrace{e_0 e_i^T}_{\text{a scalar}}. \tag{16}$$

Notably, if all $e_i$ are normalized, we should have $\arg\max_{e_i} e_i e_0^T = e_0$. Therefore, we can find this $e_i$ by simply finding the one that maximizes the norm in Eq. 16:

$$e_0 = \arg\max_{e_i} \|(\beta_k^+ - \beta_k^-)e_i^T\|_2. \tag{17}$$

## A.2  EXTENSION TO MULTIPLE-WORDS PROMPTS

In general cases, prompts consist of multiple words, making the inversion process tricky. In these cases, $e_0$ may have the shape [W, N] for SD, where W is the length of the prompts. (In fact, due to the presence of position embedding, cases with different words actually all have $W = 77$, where 77 is the maximum length for input prompts). This results in $e_0 e_i^T$ not being a scalar anymore in Eq. 16. Its shape is $[W, W]$. Therefore, we cannot obtain $e_0$ by simply computing the norm. In fact, Eq. 15 shows that the row space of $\beta_k^+ - \beta_k^-$ is span$\{e_{0,1}, e_{0,2}, \cdots, e_{0,77}\}$. Here, we can use a transformation where we perform Principal Component Analysis (PCA) decomposition (Abdi & Williams, 2010) on $\beta_k^+ - \beta_k^-$. In other words, we approximate it using a rank-one matrix:

$PCA(\beta_k^+ - \beta_k^-) \approx \overline{L_k}\overline{e_0}$, where $\overline{e_0}$ has the shape [1, N]. This can also be regarded as finding the main vector direction within the space $span\{e_{0,1}, e_{0,2}, \cdots, e_{0,77}\}$. With this decomposition, we then have:

$$e_0 \approx \arg\max_{e_i} \|(PCA(\beta_k^+ - \beta_k^-))e_i^T\|_2 \tag{18}$$

### A.3 Extension to More Optimizers and Approximation During Training with Multiple Linear Matrices

In more general cases, Adam is usually used instead of SGD, resulting in Eq. 15 not holding exactly but with some error. Moreover, if we use some adaptors for fine-tuning, such equations also have some error as we are trying to use a low-rank matrix to approximate a higher-rank matrix. Therefore, the row space for $\beta_k^+ - \beta_k^-$ may now be $span\{e_{0,1} + \epsilon_{k,1}, e_{0,2} + \epsilon_{k,2}, \cdots, e_{0,77} + \epsilon_{k,77}\}$, where $\epsilon$ is a small error term.

We can incorporate more information to reduce the effect brought by the error term. In practical scenarios, we have many linear projection layers that accept the same text embedding $e_0$ as input. For example, in SD, we may have about 20 such layers. Therefore, we can concatenate all of them together and try to find a decomposition considering them all. In other words, find a main vector for all matrices within the space $span\{e_{0,1}, e_{0,2}, \cdots, e_{0,77}, \epsilon_{1,1}, \epsilon_{1,2}, \cdots, \epsilon_{1,77}, \cdots, \epsilon_{K,1}, \epsilon_{K,2}, \cdots, \epsilon_{K,77}\}$, which corresponds to $PCA([(\beta_0^+ - \beta_0^-), (\beta_1^+ - \beta_1^-), \cdots, (\beta_K^+ - \beta_K^-)])$.

However, in practice, we find that such PCA may fail in most cases, suffering from a performance drop. This may be due to the fact that the error term $\epsilon$ is not necessarily insignificant. Therefore, we perform PCA with $\beta^+$ and $\beta^-$ respectively. Then we compute the difference between them, i.e., $PCA(\beta^+) - PCA(\beta^-)$ instead of performing PCA directly on, i.e., $PCA(\beta^+ - \beta^-)$:

$$e_0 \approx \arg\max_{e_i} F(e_i) = \arg\max_{e_i} \|(PCA[\beta_0^+, \beta_1^+, \cdots, \beta_K^+] - PCA[\beta_0^-, \beta_1^-, \cdots, \beta_K^-])e_i^T\|_2. \tag{19}$$

The intuition behind applying PCA to the row space is that it extracts the most significant signals from the training prompts. For a pretrained model $\beta^-$, such signals represents the principal vector for real-world prompts. For a fine-tuned model $\beta^+$, such signal should also highly align with real prompts, but with some information about fine-tuned captions. Using PCA first in Eq. 19 can prevent extracted embedding from diverging into unrelated or complex signals and staying within the embedding space of real-world prompts.

### A.4 Optimization

Optimization over Eq. 19 can indeed derive some prompt embeddings, but these may not be related to any real prompts. So the real optimization target should be finding a input prompt $c_0$:

$$c_0 \approx \arg\max_{c_i} F(E(c_i)), \tag{20}$$

where $E(\cdot)$ is the frozen text feature embedding function. The challenge is that the prompt space is discrete, which turns the optimization problem into a hard-prompt finding task. To overcome this, we adopt the technique from Wen et al. (2024): during optimization, we project the current embedding onto a real prompt and use the gradient of this real prompt's embedding to update our given embedding. Algorithm 1 illustrates this framework.

### A.5 Experiment Result

We conduct experiments on four classes of the WikiArt dataset, where the DMs are fine-tuned using DreamBooth. Table 5 presents comparison between the original training captions and the captions extracted by our method. The results demonstrate that representative information can be extracted

---

**Algorithm 1** Hard Prompt Extraction on Fine-tuned Caption for Linear Layers

---

**Input:** Pre-trained Linear parameters $\beta^-$, fine-tuned Linear parameters $\beta^+$, input caption $c_0$, text encoder $E$, optimization iterations $N$, learning rate $\alpha$, projection function $E^{-1}$.
**Output:** Extracted caption $\hat{c}$.
Initialize embedding $\hat{e}$ with random prompts: $\hat{e} = E(c)$.
**for** $i = 0$ **to** $N - 1$ **do**
    $\hat{c} = E^{-1}(\hat{e})$
    Find gradient $\delta = \nabla_{E(\hat{c})} F(E(\hat{c}))$ based on this $\hat{c}$, $\beta^+$ and $\beta^-$ using Eq. 19
    Update $\hat{e} \rightarrow \hat{e} + \alpha\delta$
**end for**
$\hat{c} \rightarrow E^{-1}(\hat{e})$

---

Table 5: Experimental results of our caption extraction algorithm using the L2-PGD attack with 1000 iterations starting from a random prompt. This extraction costs approximately 1.5 to 2 minutes per sample on a single A100 GPU. The results demonstrate that key information, such as the artist's name, can largely be inferred. Correctly inferred parts are shown in **bold**.

| Training Caption | Extracted Caption ($\leq$ 3 words) | Extracted Caption ($\leq$7 words) |
|---|---|---|
| art style of Post Impressionism vincent | attributed **impressionism vincent** | plein **impressionism vincent** demonstrating! fantastic :)) |
| art style of Fauvism henri matisse | **henri** paintings **matisse** | donneinarte hemingway **henri matisse** throughout fineart paintings |
| art style of High Renaissance leonardo da vinci | **leonardo** confident paintings | **leonardo** onda elengrembrandt pre picasso artwork |
| art style of Impressionism claude monet | suggestions **impressionism monet** | cassini gustave **monet impressionist monet** etosuggesti |

to some extent, indicating a leakage of training captions information based on the DMs before and after fine-tuning.

Experiment results under longer training prompt are discussed in Sec. J.

## B COMPUTATIONAL COSTS ANALYSIS

Our method, as detailed in Sec. 4.1 and 4.2, employs guidance between two distinct models at each generation step without slowing down the generation speed compared to traditional CFG (which also requires guidance by forwarding the main UNet in DMs twice). However, GPU memory costs increase due to the need to load both the pretrained and fine-tuned models. In Tab. 6, we present a demo experiment comparing the computational costs of FineXtract and CFG, using SD (v1.4) fine-tuned with DreamBooth and LoRA on the WikiArt dataset. The batch size is fixed at 5, and all experiments are conducted on a single A100 GPU.

Table 6: Generation GPU memory costs and time costs for per-image generation using FineXtract and CFG.

| Metrics and Settings | DreamBooth | | LoRA | |
|---|---|---|---|---|
| | Memory Costs (MB) | Time Costs (s) | Memory Costs (MB) | Time Costs (s) |
| FineXtract | 15812 | 3.4 | 15894 | 3.6 |
| CFG | 12096 | 3.4 | 12226 | 3.6 |

## C RESULTS WITH NO PROMPTS PROVIDED

We also explore how unconditional generation can be used to extract training images. We conduct experiments on four classes of the WikiArt dataset, using various prompts and generating images with an empty prompt. The results, shown in Fig. 8, demonstrate that FineXtract improves performance compared to the baseline (since no conditional information is available, CFG cannot be applied, so the baseline corresponds to the $w' = 1$ case in FineXtract). However, the AS is significantly lower than when a caption is provided, leading to far fewer successful extractions.

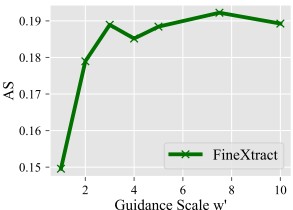

Figure 8: Experiment result when no training prompt provided.

Table 7: Information about subsets of the checkpoints used for real-world experiments.

| Model Name | SD Version | Fine-tuning Methods | # of Training Image |
|---|---|---|---|
| sd-dreambooth-library/mau-cat | SD (V1.5) | DreamBooth | 5 |
| sd-dreambooth-library/the-witcher-game-ciri | SD (V1.5) | DreamBooth | 5 |
| sd-dreambooth-library/mr-potato-head | SD (V1.5) | DreamBooth | 6 |
| Norod78/SDXL-YarnArtStyle-LoRA | SDXL (V1.0) | Lora | 14 |
| Norod78/pokirl-sdxl | SDXL (V1.0) | Lora | 22 |
| Norod78/SDXL-PringlesTube-Lora | SDXL (V1.0) | Lora | 138 |

## D    ABALTION STUDY OF $w'$ UNDER LoRA FINE-TUNING

In LoRA, we find that the suitable $w'$ for CFG and FineXtract differs, which are 3.0 and 5.0, respectively, as shown in Fig. 9. We experiment on 4 classes of WikiArt dataset fine-tuning on SD (V1.4) with LoRA.

## E    FINETUNING DETAILS

The details of the parameters in the fine-tuning methods are presented below. We use $N_0$ to represent the number of images utilized for training. Our setting mostly follow the original training setting in few-shot fine-tuning methods (Ruiz et al., 2023).

**Dreambooth**: We use the training script provided by Diffusers[6]. Only the U-Net is fine-tuned during the training process. By default, the number of training steps is set to $200 \times N_0$, with a learning rate of $2 \times 10^{-6}$. The batch size is set to 1. We set the prior loss weight as 0.0 for simplification. For the WikiArt dataset, the instance prompt is "art style of [class name]", where [class name] is the name of the artist such as "Fauvism henri matisse". For the Dreambooth dataset,

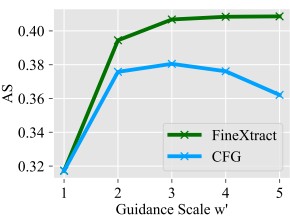

Figure 9: Ablation Study on hype-parameters $w'$.

the instance prompt is "a [class name]" where [class name] is the class of the object, such as "dog".

**LoRA**: We use the training script provided by Diffusers[7]. All default parameters remain consistent with the case in Dreambooth (No Prior), with the exception of the learning rate, which is adjusted to $1 \times 10^{-4}$. The rank is fixed to 64 to ensure the fine-tuning process capture fine-grained details of training samples. The prompts used are the same as the case in DreamBooth.

## F    REAL-WORLD EXPERIMENTS SETUP

We randomly selected 10 fine-tuned DM checkpoints from Hugging Face, which include those fine-tuned from SD (V1.5) and SDXL (V1.0) using DreamBooth and LoRA. The number of training images ranges from 5 to 138. We compute the AS and A-ESR for each checkpoint and average the results, which correspond to those presented in Tab. 4. Details of these checkpoints, all with permissive licenses, are provided in Tab. 7. These checkpoints are available on Hugging Face for result reproduction. We sincerely appreciate the checkpoint creators for sharing their work to support both research and practical applications.

## G    EXPERIMENT ON MIXTURE OF DATASET

### G.1    MIXTURE OF DREAMBOOTH AND WIKIART DATASETS

We constructed a new dataset with 10 classes, each containing 5 images from DreamBooth and 5 from WikiArt. As reported in Tab. 8, this mixture led to mutual decreases in the fine-tuned model's fidelity (measured by DINO score (Ruiz et al., 2023)), image quality (measured by CLIP-IQA (Wang

---

[6]https://github.com/huggingface/diffusers/blob/main/examples/dreambooth/train_dreambooth.py
[7]https://github.com/huggingface/diffusers/blob/main/examples/dreambooth/train_dreambooth_lora.py

Table 8: Comparison of the performance of FineXtract and the baseline in fine-tuning data extraction under separated and mixed data scenarios using DreamBooth and WikiArt datasets. We observe mixed data makes extraction more challenging while also making it harder for diffusion models to learn the given data, as evidenced by lower fidelity (DINO) and reduced image quality (CLIP-IQA).

| Dataset | Extraction Method | AS↑ | A-ESR$_{0.6}$↑ | DINO ↑ | Clip-IQA ↑ |
|---|---|---|---|---|---|
| Separated Data | CFG+Clustering | 0.525 | 0.45 | 0.533 | 0.697 |
| | FineXtract | **0.572** | **0.55** | | |
| Mixed Data | CFG+Clustering | 0.457 | 0.08 | 0.266 | 0.447 |
| | FineXtract | **0.480** | **0.18** | | |

Table 9: Comparison of the performance FineXtract and baseline in fine-tuning data extraction under dataset mixed from different number of classes of WikiArt dataset. Mixed data with different classes makes model learning poorer, causing lower fidelity (DINO) and image quality (Clip-IQA), which in turn makes extraction more challenging.

| Mixed number of Artists Per Class | Extraction Method | AS↑ | A-ESR$_{0.6}$↑ | DINO ↑ | Clip-IQA ↑ |
|---|---|---|---|---|---|
| 1 Artist Per Class | CFG+Clustering | 0.396 | 0.11 | 0.458 | 0.525 |
| | FineXtract | **0.449** | **0.22** | | |
| 2 Artist Per Class | CFG+Clustering | 0.390 | **0.20** | 0.387 | 0.441 |
| | FineXtract | **0.436** | **0.20** | | |
| 5 Artist Per Class | CFG+Clustering | 0.353 | 0.15 | 0.343 | 0.478 |
| | FineXtract | **0.388** | **0.23** | | |

et al., 2023)), and extraction rate. Therefore, extracting training images from this model becomes more challenging. Nonetheless, FineXtract still significantly outperforms the baseline despite the fact that its performance partially drops compared to the original cases, further verifying its generality and effectiveness.

### G.2 MIXTURE OF DIFFERENT STYLES IN WIKIART DATASETS

We conducted experiments with an increasing number of classes and found that while it reduces our method's performance, we still significantly outperform the baseline. As shown in Tab. 9, we observe that as the number of styles increases, the extraction success rate decreases. Additionally, we evaluated the model's ability to learn the input distribution and found that a higher number of classes leads to lower fidelity (DINO) and image quality (Clip-IQA), reflecting the model's difficulty in learning the fine-tuning data distribution accurately.

### H ABLATION STUDY ON MODEL GUIDANCE

We provide further discussion by comparing model guidance using conditional DM, CFG, and a combination of CFG and model guidance (with $k = 0$). As shown in Fig. 10, model guidance using conditional DM achieves close performance to the combination method and significantly outperforms CFG alone, suggesting model guidance's dominance and a potential misalignment with CFG, affecting the parameter $k$.

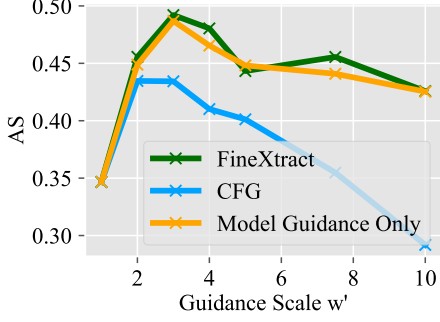

Figure 10: Ablation study on different $w'$ dealing with model guidance, CFG, and FineXtract. We experiment on 4 classes of WikiArt dataset.

### I MORE ASSESSMENTS FOR PRE-PROCESSING DEFENSE

We attempted to quantify usability using a no-reference image quality score and the image fidelity measurements. We follow previous works (Ruiz

Table 10: Image fidelity and quality for checkpoints under different defenses. DINO measures image fidelity with respect to the training dataset, and thus it is not applicable when dealing with source images. We use "N.A." to denote this scenario.

| Defenses | Source Images | | Generated Images | |
|---|---|---|---|---|
| | DINO↑ | Clip-IQA↑ | DINO↑ | Clip-IQA↑ |
| No Defense | N.A. | 0.568 | 0.458 | 0.525 |
| CutOut | N.A. | 0.507 | 0.460 | 0.487 |
| RandAugment | N.A. | 0.522 | 0.435 | 0.479 |

et al., 2023), using DINO for image fidelity measurements. For image quality measurements, we use Clip-IQA (Wang et al., 2023). As shown in Tab. 10, we found that the CutOut and RandAugment degraded Clip-IQA by 6.1% and 4.6% in the original dataset, and generated images experienced around 3.8% and 4.6% degradation. These extent are close in the sense that the degration brought by the transformation largely preserves in the generation process. For RandAugment, the image fidelity also largely degrades. This suggests that the preprocessing steps applied to the images (removing a square section or adding high contrast to the input images) largely persist in the generated output images, hindering extraction methods from obtaining high-quality images while sacrificing generation quality of diffusion models themselves.

In summary, there exists a **trade-off between image quality and defensive effects** for these preprocessing methods. We leave the accurate modeling of this trade-off and further improvement as an interesting future work.

## J    EXPERIMENT ON EXTRACTING PROMPTS UNDER DIFFERENT PROMPT LENGTH $W$

We further experiment on different scenarios involving fine-tuning with longer training prompts. Specifically, we use GPT-4 to expand the original prompts into longer versions by adding information related to the author while avoiding overly specific details to prevent mismatches with the input image. Tab. 11 shows our extraction results. With longer text, extracting detailed information becomes more challenging. However, our algorithm can still identify specific details, such as the artist's name, even with extended text. When the text length continue to increase, certain information becomes harder to extract, though such cases are rare in few-shot fine-tuning.

Table 11: Experimental results of our caption extraction algorithm using the L2-PGD attack with 1000 iterations starting from a random prompt. This extraction costs approximately 1.5 to 2 minutes per sample on a single A100 GPU. The results demonstrate that key information, such as the artist's name, can largely be inferred when the training caption is not too long. Correctly inferred key-word parts are shown in **bold**. Related information extracted is shown in *italic*.

**Extended Prompt**

| Training Caption | Extracted Caption ($\leq$ 7 words) |
|---|---|
| An insight into the Post-Impressionist style with expressive, emotional brushwork and rich colors, inspired by the unique techniques of Vincent van Gogh. | **vangogh impressionism** class **impressionist** paintings acqu |
| Exploring the art style of Fauvism, characterized by vivid, bold colors and dynamic brushstrokes, as seen in the works of Henri Matisse. | **henri matisse** whose explores whose colourful stures |
| An exploration of the High Renaissance style, marked by balance, realism, and anatomical accuracy, capturing the essence of Leonardo da Vinci's art. | cws inaccurate anatomy **leonardo** deus aron onda |
| A look at Impressionism, noted for soft, light-filled scenes and gentle brushstrokes, inspired by the atmospheric beauty in Claude Monet's paintings. | **impressionism** wgleagues **monet** het commence |

**Excessively Extended Prompt**

| Training Caption | Extracted Caption ($\leq$ 7 words) |
|---|---|
| A deeper look into Post-Impressionism, emphasizing Vincent van Gogh's expressive, emotional brushwork and contrasting colors. His techniques convey a personal, intense view of the world, moving beyond the lighter tones of Impressionism. | **post impressionism vincent gogh** pcdimonet opposite **vangogh** |
| In-depth exploration of Fauvism's vivid colors and bold, expressive brushstrokes that bring emotions to life, heavily inspired by the renowned techniques of Henri Matisse. His approach captures the vibrant essence of Fauvism, using striking colors to convey feeling. | congratulations adrian edgar forum awaits muromain destination |
| An insightful exploration into the High Renaissance, highlighting the balance, anatomical realism, and idealism of Leonardo da Vinci's creations. This style embodies the Renaissance's focus on perfection, with meticulously detailed compositions and harmonious proportions. | retains vehicles marriott alcatraspberries tuesdaythoughts shua |
| A detailed depiction of Impressionism, capturing transient beauty with soft brushstrokes and delicate light, influenced by Claude Monet's signature style. His work portrays nature in harmonious, gentle colors that evoke calmness. | **impressionism** excellent *depiction* effectively arranged unexrepresented explained |

