# OpenReview forum: "Revealing the Unseen: Guiding Personalized Diffusion Models to Expose Training Data"
_ICLR.cc/2025/Conference — Submitted to ICLR 2025_

### Official Review · Reviewer_zPvq · 2024-11-01

**Soundness:** 3
**Presentation:** 2
**Contribution:** 3
**Rating:** 6
**Confidence:** 3

**Summary:**

The paper proposed a framework FineXtract, which exploits the transition from the pre-trained DM distribution to the fine-tuning data distribution to accurately guide the generation process to the high-probability region of the fine-tuning data distribution, thereby achieving successful data extraction. Experiments on multiple datasets and real-world checkpoints, highlight the potential risks of data leakage and provide strong evidence for copyright infringement.

**Strengths:**

1. This paper is well-written.
2. This framework can be applied to both unconditional and conditional DMs.
3. The result is significant, highlighting the potential risks of data leakage.

**Weaknesses:**

1. In Sec. 5.2, the performance of baselines under various $N$ and $N_0$ deserves further discussion.
2. The reason why Correction Term Scale $k$ performs better in the negative case needs further analysis, which is inconsistent with its motivation.
3. It is worrying whether using PCA to extract important signals from multiple-words prompts is feasible when $W$ is large.
4. Some symbol errors, for example, the second $\epsilon_{0,77}$ in Appendix A.3 should be $\epsilon_{1,77}$.

**Questions:**

This paper is both interesting and innovative. However, there are some weaknesses that need to be improved. Please refer to Weaknesses.

---

> ### Author Response · Authors · 2024-11-21
> **Official Comments by Authors**
>
> We thank the reviewer for valuable feedback, and would like to address your concerns as belows:
>
> For Weakness1:
>
> > baselines under various $\rm{N}$ and $\rm{N_0}$ deserves further discussion.
>
> We updated the results for different $\rm{N}$ and $\rm{N_0}$ values compared to the baseline in Fig. 4(a) and Fig. 4(b), where FineXtract consistently outperforms baseline under different $\rm{N}$ and $\rm{N_0}$.
>
>
> For Weakness2:
>
> > why Correction Term Scale  performs better in the negative case needs further analysis
>
> Our method in Sec. 4 is a parametric approximation. We provide further discussion by comparing model guidance using conditional DM, CFG, and a combination of CFG and model guidance (with $k=0$). As shown in Fig.10 in Appendix Sec.H , model guidance using conditional DM achieves close performance to the combination method and significantly outperforms CFG alone, suggesting model guidance’s dominance and a potential misalignment with CFG, affecting the parameter $k$.
>
>
> For Weakness3:
>
> > It is worrying whether using PCA to extract important signals from multiple-words prompts is feasible when $\rm{W}$  is large.
>
> Increasing $\rm{W}$ indeed makes caption extraction more difficult. We add some experiments on larger $\rm{W}$ in Appendix Sec. J, where we use GPT-4 to generate extended artist style descriptions. Our results show that keywords, such as the artist’s name, remain extractable as long as the training prompt is not excessively long. This indicates that some unique words in the fine-tuning process still lead to partial information leakage.
>
>
> For Weakness4: symbol errors in Appendix A.3.
>
> Thank you for pointing it out. We have revised it accordingly.

---

> > ### Comment · Reviewer_zPvq · 2024-11-25
> >
> > Thank you for the explanation and revision. I will keep my rating.

---

### Official Review · Reviewer_rSyw · 2024-11-02

**Soundness:** 2
**Presentation:** 3
**Contribution:** 2
**Rating:** 6
**Confidence:** 3

**Summary:**

This paper studies the data extraction problem of diffusion model, particularly focusing on the fine-tuning data. The authors use the parametric approximation of the distribution shift between the original model and fine-tuned model as guidance, to generate the fine-tuning data. Experiments across different diffusion models on various datasets and real-world checkpoints from huggingface demonstrate the effectiveness of proposed method.

**Strengths:**

1.	The focus on extracting fine-tuning data is indeed interesting. this focus reveals a new perspective on privacy concerns, which could enhance the security of diffusion models and preserve the privacy of data owners.
2.	The experiments are also conducted on checkpoints from real-world platform, i.e., huggingface, demonstrating the practical effectiveness of the proposed method.
3.	The paper is generally well-written, with a clear structure that is easy to follow.

**Weaknesses:**

1.	The performance of the proposed method keeps decreasing with the growth of training data. Will the growth of class number have the same effect? How can this issue be mitigated in practice, especially given the vast volume of training data used for industry diffusion models?
2.	The performance under LoRA fine-tuning is noticeably worse. Does this suggest that the proposed method is less effective for parameter-efficient tuning?
3.	The effectiveness of the proposed method is significantly diminished when being attacked. The authors state that "transformations render ... images largely unusable." Could you provide statistics on the extent of unusability? To what degree does the attacker lose model utility to achieve the attack performance reported in Table 3?

**Questions:**

Please refer to Weaknesses.

---

> ### Author Response · Authors · 2024-11-21
> **Official Comments by Authors**
>
> We thank the reviewer for valuable feedback, and would like to address your concerns as belows:
>
>
> 1. For Weakness1:
>
> > Will the growth of class number have the same effect (Performance drop in extraction)?
>
> Yes, increasing the number of classes makes it harder for diffusion models to learn the fine-tuned data distribution, which in turn results in more challenging extraction and an observed performance drop. Specifically:
>
> We conducted experiments with an increasing number of classes and found that while it reduces our method’s performance, we still significantly outperform the baseline. As shown in Sec.G.2, we observe that as the number of styles increases, the extraction success rate decreases. Additionally, we evaluated the model's ability to learn the input distribution and found that a higher number of classes leads to lower fidelity (DINO) and image quality (Clip-IQA), reflecting the model's difficulty in learning the fine-tuning data distribution accurately.
>
> > How can this issue be mitigated in practice
>
>
> To further enhance extraction under these scenarios, utilizing accessible public datasets may be helpful. For example, extracting checkpoints trained in a specific artist's style could be improved by leveraging reference images of similar styles. This direction is also part of our future work.
>
> 2. For Weakness2:
>
> >  Does this (LoRA with poorer extraction performance) suggest that the proposed method is less effective for parameter-efficient tuning?
>
> No. Our FineXtract appears to perform worse on parameter-effecient tuning methods like LoRA because the fine-tuned models learn worse on the data distribution. For evidence, we performed an ablation study on different training iterations for LoRA with other training parameters for LoRA and DreamBooth following previous work [1,2].
>
> We found that performance issues do not stem from LoRA itself but from how well the model learns the fine-tuning data distribution. As shown in the table below, LoRA also achieves a higher extraction success rate than DreamBooth when LoRA’s training iterations are extended. So the key factor for extraction success is more about how well a DM learns the fine-tuning data distribution, instead of the fine-tuning method itself.
>
>
> | Dataset            | DreamBooth (200 N₀)       |            | LoRA (200 N₀)           |            | LoRA (300 N₀)           |            | LoRA (400 N₀)           |            |
> |--------------------|---------------------------|------------|--------------------------|------------|--------------------------|------------|--------------------------|------------|
> |                    | AS ↑                     | A-ESR₀.₆ ↑ | AS ↑                    | A-ESR₀.₆ ↑ | AS ↑                    | A-ESR₀.₆ ↑ | AS ↑                    | A-ESR₀.₆ ↑ |
> | **CFG+Clustering** | 0.396                    | 0.11       | 0.380                   | 0.03       | 0.418                   | 0.15       | 0.522                   | 0.25       |
> | **FineXtract**     | **0.449**                | **0.22**   | **0.405**               | **0.10**   | **0.445**               | **0.18**   | **0.554**               | **0.48**   |
>
>
>  3. For Weakness3:
>
> >  Could you provide statistics on the extent of unusability? To what degree does the attacker lose model utility to achieve the attack performance reported in Table 3?
>
> Measuring usability precisely is challenging, but we attempted to quantify it using a no-reference image quality score and the image fidelity measurements. We follow previous works [1], using DINO for image fidelity measurements. For image quality measurements, we use Clip-IQA [2]. We update some experiment results in Appendix Sec. I and find that the CutOut and RandAugment degraded Clip-IQA by 6.1\% and 4.6\% in the original dataset, and generated images experienced around 3.8\% and 4.6\% degradation. These extent are close in the sense that the degration brought by the transformation largely preserves in the generation process. For RandAugment, the image fidelity also largely degrades. This suggests that the preprocessing steps applied to the images (removing a square section or adding high contrast to the input images) largely persist in the generated output images, hindering extraction methods from obtaining high-quality images while sacrificing generation quality of diffusion models themselves.
>
> In summary, there exists a  **trade-off between image quality and defensive effects**  for these preprocessing methods.
> We leave the accurate modeling of this trade-off and further improvement as an interesting future work.
>
> [1] Nataniel Ruiz, Yuanzhen Li, Varun Jampani, Yael Pritch, Michael Rubinstein, and Kfir Aberman.
> DreamBooth: Fine Tuning Text-to-Image Diffusion Models for Subject-Driven Generation. In
> CVPR, 2023
>
> [2] Jianyi Wang, Kelvin CK Chan, and Chen Change Loy. Exploring Clip for Assessing the Look and
> Feel of Images. In AAAI, 2023.

---

> > ### Comment · Reviewer_rSyw · 2024-11-25
> >
> > Thank you for the response. I would like to maintain my rating.

---

### Official Review · Reviewer_EEJp · 2024-11-04

**Soundness:** 2
**Presentation:** 3
**Contribution:** 3
**Rating:** 5
**Confidence:** 3

**Summary:**

This paper introduces a novel framework FineXtract to extract fine-tuning data from fine-tuned diffusion models' checkpoints. The authors approximate the learned distribution during the fine-tuning process of diffusion models and use it to guide the generation process toward high-probability regions of the fine-tuned data distribution. Besides, a clustering algorithm is proposed to extract images visually close to fine-tuning datasets. Experiments result on fine-tuned checkpoints on various datasets, various diffusion models verify the effectiveness of the proposed FineXtract.

**Strengths:**

1. The paper is overall clear written and easy to follow.
2. The paper focuses on extracting fine-tuning data from diffusion models' checkpoints. This research topic has not been paid attention before.
3. The code is available.

**Weaknesses:**

1. The experiment of the paper is not solid enough, the paper needs more experiments to verify the proposed method works. The paper only chooses "Direct Text2img+Clustering" and "Classifier Free Guidance + CLustering" as baselines. I think these two methods are only ablation counterparts. It is better to compare the proposed method with other relative works on extracting training/finetuning data.
2. The proposed method seems sensitive to the guidance scale $w'$ and correction term $k$. How to decide the hyper-parameters in practice might be challenging.
3. I am somewhat skeptical about the necessity of developing a dedicated method specifically for extracting images from the fine-tuning phase. It seems feasible to simply apply existing methods for extracting training images directly on the fine-tuned checkpoint, then filter out the results that overlap with images extracted from the pretrained model.

**Questions:**

Please see "Weaknesses".

---

> ### Author Response · Authors · 2024-11-21
> **Official Comments by Authors**
>
> We thank the reviewer for valuable feedback, and would like to address your concerns as belows:
>
>
> 1. For Weakness1:
>
> > The paper only chooses 'Direct Text2img+Clustering' and 'Classifier Free Guidance + Clustering' as baselines. I think these two methods are only ablation counterparts.
>
> These two baselines are not ablation counterparts. To the best of our knowledge, prior work on data extraction from diffusion models [1,2,3] primarily discusses methods related to direct text-to-image extraction [1] and CFG-based approaches [2,3]. However, no existing methods specifically target the extraction of fine-tuning datasets. Therefore, the baselines listed here represent all the available methods we could identify from prior work related to diffusion model extraction.
>
>
> 2. For Weakness2:
>
> >  The proposed method seems sensitive to the guidance scale $w'$  and correction term $k$ . How to decide the hyper-parameters in practice might be challenging.
>
> Firstly, the improvement from our proposed FineXtract is not sensitive to the hyperparameters. As shown in Tab. 5, the best performance of the baseline method is $AS\approx 0.44$ with only $w' = 2$ or $3$. However, FineXtract performs better under a large scale of guidance scale $w'$  and correction term $k$, i.e. $w'\in [2, 10]$ and all $k$s.
>
> Secondly, we do not heavily rely on hyperparameter tuning. Concretely, our approach involves searching on 4 classes of WikiArt and applying the resulting hyperparameters universally across various scenarios, including the DreamBooth dataset, all 10 classes of WikiArt, and real-world checkpoints (where most training details remain inaccessible).  Developing more tailored hyperparameters for specific scenarios should further enhance performance.
>
> 3. For Weakness3:
>
> > It seems feasible to simply apply existing methods for extracting training images directly on the fine-tuned checkpoint, then filter out the results that overlap with images extracted from the pretrained model.
>
> In principle, the overlapping from extracted train images from fine-tuned checkpoint and pretrained model can only be the images from the pretrained dataset, while our task is to extract training data used during the fine-tuning process, as clearly stated in Page 1. In practice, previous works on extracting pretrained diffusion models [1] successfully extracted fewer than 100 images, none of which overlaps with the fine-tuning dataset and this number is much lower than the size of its training dataset size (more than 200K). On the other hand, our main results in Tab. 1 demonstrate that directly applying existing methods (i.e., the shown baselines) performs significantly worse than our proposed method on the fine-tuned data extraction task, only achieving around half of our extraction success rates in many cases.
>
> [1] Nicolas Carlini, Jamie Hayes, Milad Nasr, Matthew Jagielski, Vikash Sehwag, Florian Tramer, Borja
> Balle, Daphne Ippolito, and Eric Wallace. Extracting training data from diffusion models. In 32nd
> USENIX Security Symposium (USENIX Security 23), pp. 5253–5270, 2023
>
> [2] Gowthami Somepalli, Vasu Singla, Micah Goldblum, Jonas Geiping, and Tom Goldstein. Diffusion
> art or digital forgery? investigating data replication in diffusion models. In Proceedings of the
> IEEE/CVF Conference on Computer Vision and Pattern Recognition, pp. 6048–6058, 2023
>
> [3] Gowthami Somepalli, Vasu Singla, Micah Goldblum, Jonas Geiping, and Tom Goldstein. Under
> standing and mitigating copying in diffusion models. Advances in Neural Information Processing
> Systems, 36:47783–47803, 2023b.

---

> > ### Comment · Reviewer_EEJp · 2024-11-27
> >
> > Thanks for the authors' response. After reading other reviews and revisiting the paper, I would like to maintain my rating.

---

### Official Review · Reviewer_U46r · 2024-11-04

**Soundness:** 3
**Presentation:** 3
**Contribution:** 3
**Rating:** 5
**Confidence:** 3

**Summary:**

This paper proposes a model guidance-based approach to fine-tuning data recovery, where it leverages the denoising prediction from pre-trained models $\epsilon_{\theta}$ to extrapolate the predictions from fine-tuned models $\epsilon_{\theta^{'}}$. The resulting denoiser can be used to sample data from a distribution similar to the fine-tuned data distribution $q$. The authors further extend this to text guidance diffusion models. After denoising steps, a clustering algorithm was applied to further improve the extraction accuracy. The experimentations demonstrate improved average similarity and average extraction accuracy of extracted images compared to text-to-image and CFG with clustering as baselines. Further ablation study was conducted to understand the impact of training images $N_0$ and generated images $N$, as well as model hyperparameters such as guidance scale $w'$ and correction term scale $k$. Results on possible defenses against the proposed method were also presented.

**Strengths:**

- The paper is well-written with clear presentations. For example, the method sections of model guidance and text-guidance extension are presented in a way that is easy to follow, and further details on caption extraction are elaborated in the appendix. The figures and tables are quite easy to parse and get the main results.
- The experimentations are quite comprehensive, with ablation studies on several important hyperparameters concerning models and datasets.
- It is also very good to present some defenses against the proposed method and discuss the implications of these results

**Weaknesses:**

The main weaknesses are the task setup and the significance of results:
- For task setup, the paper seems to address a relatively clean form of fine-tuned models, whereas in real-world settings the pre-trained models might be not always available (sometimes presented with noisy labels), and in many cases, the fine-tuned model could be a mixture of multiple fine-tuned data distributions and pre-trained models. I wonder how the proposed methods were able to consider much more realistic scenarios.
- The main experiments are conducted on a relatively small test dataset that consists of 20 artist models (10 mages per model) and 30 object models (4-6 images per model), making the significance of results hard to judge. Moreover, the improvements over the two selected baselines are noticeable (table 1). When increasing $N_{0}$, the performance drops significantly (figure 4a), the model does not work very well on fine-tuned models with a larger scale of images. These results suggest space for improvements, which are all needed from this work considering the applications of real-world problems.

**Questions:**

1. Can you provide more intuition on eq (3)? For example, besides the training iterations effects on $\lambda$, how $\lambda$ might be affected by the size of fine-tuned data and their similarities?
2. What's the performance of extraction accuracy with increasing fine-tuning data $N_{0}$?
3. Can you conduct more analyses on the impact of different combinations of DMs and training data (artistic styles vs. object). It seems their performance are quite different from table 1 and table 2.

---

> ### Author Response · Authors · 2024-11-21
> **Official Comments by Authors**
>
> We thank the reviewer for valuable feedback, and would like to address your concerns as below:
>
>
> 1. For Weakness1 & Question3:
>
> > in real-world settings the pre-trained models might be not always available ... the fine-tuned model could be a mixture of multiple fine-tuned data distributions and pre-trained models.
>
> > Can you conduct more analyses on the impact of different combinations of DMs and training data (artistic styles vs. object)?
>
> From our analysis of real-world checkpoints available online, such as those on Civitai and Hugging Face, pretrained model checkpoints are typically clearly stated or provided with direct download links. Meanwhile, parameter-efficient fine-tuned methods such as LoRA commonly require alignment with the exact pretrained model to correctly load
> an adapter, making pretrained model checkpoints even more accessible.
>
>
> As for the training data distribution for few-shot fine-tuning, based on previous work [1] and the checkpoints we examined online, a specific style or object is typically associated with learning from data corresponding to a single class. Nonetheless, we fully agree with the reviewers that extending to more diverse scenarios would indeed be valuable. Specifically:
>
>
> **Mixture of data:** We constructed a new dataset with 10 classes, each containing 5 images from DreamBooth and 5 from WikiArt:
> | Dataset         | Extraction Method | AS ↑   | A-ESR₀.₆ ↑ | DINO ↑   | Clip-IQA ↑ |
> |-----------------|-------------------|--------|-------------|----------|------------|
> | **Separated Data** | CFG+Clustering  | 0.525  | 0.45        | 0.533    | 0.697      |
> |                 | FineXtract        | **0.572** | **0.55**   |          |            |
> | **Mixed Data**    | CFG+Clustering  | 0.457  | 0.08        | 0.266    | 0.447      |
> |                 | FineXtract        | **0.480** | **0.18**   |          |            |
>
> More details about the experiment are available in Appendix Sec. G.1 in the revised paper.
> We can observe that this mixture led to mutual decreases in the fine-tuned model's fidelity (measured by DINO score [1]), image quality (measured by CLIP-IQA [2]), and extraction rate.  Therefore, extracting training images from this model becomes more challenging. Nonetheless, FineXtract still significantly outperforms the baseline despite the fact that its performance partially drops compared to the original cases, further verifying its generality and effectiveness.
>
> **About the mixture of pretrained checkpoints:** To the best of our knowledge, we are not aware of any prior work on data extraction or, more broadly, privacy-related topics in diffusion models that discusses the combination of different pretrained checkpoints. We are unsure about the specific mixture the reviewer is referring to and would greatly appreciate further clarification on this point.
>
> 2. For Weakness2 & Question2:
>
> > When increasing $\rm{N_0}$, the performance drops significantly (figure 4a), the model does not work very well on fine-tuned models with a larger scale of images. These results suggest space for improvements, which are all needed from this work considering the applications of real-world problems.
>
> > What's the performance of extraction accuracy with increasing fine-tuning data $\rm{N_0}$?
>
> In practice, the fine-tuning dataset size $N_0$ is usually fewer than $50$ to learn a more specific style or object.
> Under this scale, our FineXtract approach performs well, as supported by results on randomly chosen real-world checkpoints available from HuggingFace with different $N_0$ presented in Tab. 4 and 7.
>
>
> As $N_0$ increases, the model is less likely to memorize individual images, hence extraction indeed becomes more challenging.
> To further confirm the superiority of our FineXtract method under this more challenging scenario, we conducted evaluation across increasing $N_0$ as shown in Fig. 4(a).
> Our FineXtract method significantly outperforms existing CFG approaches, further validating its effectiveness and generality.
>
> 3. For Question1 about intuition of Eq (3):
>
> Our formulation in Eq. (3) basically assumes fine-tuning learns a mixture of pretrained model distribution and fine-tuning data distribution.
> The variable $\lambda$ $\in [0, 1]$ reflects how well the fine-tuning model memorizes the fine-tuned data distribution. If the model exactly memorizes the fine-tuning data distribution, potentially completely overfitting on it, $\lambda$ would be 1.
> Thus, any factors increasing memorization ability, such as larger training iterations, smaller fine-tuning datasets, or reduced complexity in the dataset, should increase $\lambda$.
>
> [1] Nataniel Ruiz, Yuanzhen Li, Varun Jampani, Yael Pritch, Michael Rubinstein, and Kfir Aberman.
> DreamBooth: Fine Tuning Text-to-Image Diffusion Models for Subject-Driven Generation. In
> CVPR, 2023
>
> [2] Jianyi Wang, Kelvin CK Chan, and Chen Change Loy. Exploring Clip for Assessing the Look and
> Feel of Images. In AAAI, 2023.

---

> > ### Comment · Reviewer_U46r · 2024-11-27
> >
> > Thank you for the response.
> >
> > $\textbf{mixture of pre-trained checkpoints}$: To provide more context, Diffusion Soups [1] studies the effectiveness of merging pre-trained model checkpoints. In practice, Stable Diffusion WebUI (https://github.com/AUTOMATIC1111/stable-diffusion-webui), a popular tool for users to create model checkpoints and afterwards share to platforms such as Civitai and HuggingFace, offers a "Checkpoint Merger" that allows merging of pre-trained model checkpoints. Checkpoints created by model merging have been widely appearing on these platforms such as https://huggingface.co/wikeeyang/Flux.1-Dedistilled-Mix-Tuned-fp8 and https://civitai.com/models/3450/moistmix. I would like to hear about the thoughts from the authors on how such settings can be addressed.
> >
> > $\textbf{experiements on mixture of data}$: I appreciate the extra experiments provided by the authors. Although it outperforms the compared baseline method, I'm concerned with the practicality of the proposed method in real-world applications. The huge drop in A-ESR$_{0.6}$ from 0.55 to 0.18 is far from a working solution.
> >
> >
> >
> >
> > [1] Biggs, B. et al. (2025). Diffusion Soup: Model Merging for Text-to-Image Diffusion Models. In: Leonardis, A., Ricci, E., Roth, S., Russakovsky, O., Sattler, T., Varol, G. (eds) Computer Vision – ECCV 2024. ECCV 2024. https://doi.org/10.1007/978-3-031-73036-8_15

---

> > > ### Author Response · Authors · 2024-11-28
> > > **Official Comments by Authors**
> > >
> > > Thank you for the valuable feedback. Below, we provide our responses:
> > >
> > > **About Mixture of Pre-Trained Checkpoints:**
> > >
> > > 1. We reviewed the most downloaded checkpoints on CivitAI and found that approximately 30% belong to the category of merged checkpoints. The remaining checkpoints fall within the scope of our threat model, where standard pre-trained model checkpoints are readily accessible.
> > >
> > > 2. Even for merged pre-trained checkpoints, the style transfer process often relies on publicly available merged checkpoints during fine-tuning or inference. For example, [XSMerge RealisticVisionV3 for Architectural](https://civitai.com/models/102895/xsmerge-realisticvisionv3-forarchitectural) utilizes an open-source merged checkpoint, [Realistic Vision v6.0](https://civitai.com/models/4201/realistic-vision-v60-b1?modelVersionId=130072), to enhance personalized generation capabilities for architectural designs. This demonstrates that the act of merging checkpoints does not necessarily reduce the accessibility of pre-trained checkpoints.
> > >
> > > 3. We hypothesize that the reviewer may be referring to the rest of the cases where a user privately merges two pre-trained checkpoints without disclosing the merging details and then fine-tunes the resulting model on a small dataset. In such scenarios, the attacker could still mostly identify the base models used in the merge because most open-source checkpoint licenses (e.g., CreativeML Open RAIL-M) require attribution of the source checkpoints used in the merging process. This requirement is evident in many merged checkpoints on CivitAI, where the source checkpoints are clearly specified (e.g., [Mustyle](https://civitai.com/models/985912/mustyle?modelVersionId=1104453) and [Aneurysm by ZH or Photorealistic Model Merge](https://civitai.com/models/229130/aneurysm-by-zh-or-photorealistic-model-merge?modelVersionId=1103284)). Consequently, the problem becomes an open research question about partial parameter inversion. An attacker would need to compare the fine-tuned model with potential source models to deduce the hyperparameters used during merging, potentially by measuring feature distances between models at general performance layers (e.g., layers unaffected by LoRA loading). Given that the ECCV paper [1] mentioned by the reviewer is newly published, discussions about the security, privacy, and robustness of such merging techniques are more appropriate as future work rather than issues addressed in this paper.
> > >
> > > **About Mixture of Data:**
> > >
> > > It is important to acknowledge the inherent trade-off between image quality, fidelity, and a model’s resistance to our attack. As shown in Tables 8 and 9 in Appendix Section G, mixing data from two different datasets or mixing classes within a single dataset increases resistance to extraction attacks. However, this comes at the cost of significantly degrading both image quality and fidelity (i.e., the extent to which the model captures a specific style). Consequently, while these cases appear to exhibit higher resistance to our attack, they also undermine practical usability, highlighting that FineXtract's performance is not inherently weak. Conversely, achieving high fidelity and quality with mixed data may require longer training iterations, which, in turn, increases the likelihood of successful extraction.
> > >
> > > Furthermore, even a 10–20% extraction success rate can be impactful. For instance, validating an infringement might require only one or two successfully extracted samples. Similarly, leaking sensitive information, such as personal-sensitive data, even for one image, could have severe consequences. Prior work on training data extraction for Stable Diffusion has demonstrated success rates of less than 0.01% [2], yet these rates were sufficient to highlight vulnerabilities.
> > >
> > > [1] Biggs, B. et al. (2025). Diffusion Soup: Model Merging for Text-to-Image Diffusion Models. In: Leonardis, A., Ricci, E., Roth, S., Russakovsky, O., Sattler, T., Varol, G. (eds) Computer Vision – ECCV 2024. ECCV 2024. https://doi.org/10.1007/978-3-031-73036-8_15
> > >
> > > [2] Nicolas Carlini, Jamie Hayes, Milad Nasr, Matthew Jagielski, Vikash Sehwag, Florian Tramer, Borja Balle, Daphne Ippolito, and Eric Wallace. Extracting Training Data from Diffusion Models. In *32nd USENIX Security Symposium (USENIX Security 23)*, pp. 5253–5270, 2023.

---

> > > > ### Comment · Reviewer_U46r · 2024-11-28
> > > >
> > > > Thank you for the rebuttal. I would like to maintain the original rating after considering the author's revisions and other reviewers' comments.

---

### Author Response · Authors · 2024-11-21
**Overall Comments by Authors**

We thank all reviewers for their valuable feedback and have revised the paper accordingly. The revised parts in the original text are noted in blue, while newly added sections are indicated in blue in the subsection titles of the updated draft. The revisions include:

[1] A comparison with baseline methods under different training image counts $N_0$ and generated image counts $N$ in Fig. 4.

[2] Experimental results on a mixture of WikiArt and DreamBooth datasets, as well as a mixture of different classes in WikiArt, are detailed in Appendix Sec. G.

[3] An ablation study focusing solely on model guidance, is presented in Appendix Sec. H.

[4] Quantitative assessments of pre-processing defenses, are included in Appendix Sec. I.

[5] Prompt extraction results for longer prompt lengths $W$ shown in Appendix Sec. J.

---

### Comment · Area_Chair_zARC · 2024-11-24

Dear reviewers,

Thanks for serving as a reviewer. As the discussion period comes to a close and the authors have submitted their rebuttals, I kindly ask you to take a moment to review them and provide any final comments.

If you have already updated your comments, please disregard this message.

Thank you once again for your dedication to the OpenReview process.

Best,

Area Chair

---

### Meta-Review · Area_Chair_zARC · 2024-12-20

**Metareview:**

This paper proposes a novel method called FineXtract to extract fine-tuning data from fine-tuned diffusion models' checkpoints. The authors use the approximated learned distribution of the fine-tuning process to guide DMs generation process. Experiments demonstrate their effectiveness against the baseline methods.

Strength:

1. Written is clear.

2. Experiments show their good performance against baselines.

Weaknesses:

1. Their results are not so effective when considering more classes and more training samples, which weakens the methods' contribution.

2. Evaluations are limited. Both baselines for mitigation and extraction are limited. There are a lot of papers these days discussing generating copied images or extracting images from diffusion models, e.g. [1,2] and more. The authors neglect them and make the paper's contribution uncertain.

3. In fact, extracting fine-tuning images is already the widely used setting in diffusion models. Like experiments in [3], they first fine-tune their stable diffusion and then test the image extraction as testing the whole LAION-5B is too expensive. Therefore, the author's question "Can training data be extracted from these fine-tuned DMs shared online?” has already been answered. This further weakens the contributions.

In summary, I think the paper needs more discussions on its contributions and modifications for its effectiveness before the acceptance.

[1] Detecting, Explaining, and Mitigating Memorization in Diffusion Models. ICLR 2024.
[2] Unveiling and Mitigating Memorization in Text-to-image Diffusion Models through Cross Attention. ECCV 2024.
[3] Understanding and mitigating copying in diffusion models.

**Additional Comments On Reviewer Discussion:**

The authors and reviewers discussed their attack methods' evaluation settings.

---

### Decision · Program_Chairs · 2025-01-22

Reject